# *cd*^1^ Mutation in *Drosophila* Affects Phenoxazinone Synthase Catalytic Site and Impairs Long-Term Memory

**DOI:** 10.3390/ijms232012356

**Published:** 2022-10-15

**Authors:** Aleksandr V. Zhuravlev, Polina N. Ivanova, Ksenia A. Makaveeva, Gennadii A. Zakharov, Ekaterina A. Nikitina, Elena V. Savvateeva-Popova

**Affiliations:** 1Pavlov Institute of Physiology, Russian Academy of Sciences, 199034 Saint Petersburg, Russia; 2Faculty of Biology, Herzen State Pedagogical University of Russia, 191186 Saint Petersburg, Russia; 3EPAM Systems Inc., 41 University Drive, Suite 202, Newtown, PA 18940, USA

**Keywords:** *Drosophila*, *cardinal*, 3-hydroxykynurenine, phenoxazinone synthase, courtship suppression, long-term memory, dopaminergic neurons

## Abstract

Being involved in development of Huntington’s, Parkinson’s and Alzheimer’s diseases, kynurenine pathway (KP) of tryptophan metabolism plays a significant role in modulation of neuropathology. Accumulation of a prooxidant 3-hydroxykynurenine (3-HOK) leads to oxidative stress and neuronal cell apoptosis. *Drosophila* mutant *cardinal* (*cd*^1^) with 3-HOK excess shows age-dependent neurodegeneration and short-term memory impairments, thereby presenting a model for senile dementia. Although *cd* gene for phenoxazinone synthase (PHS) catalyzing 3-HOK dimerization has been presumed to harbor the *cd*^1^ mutation, its molecular nature remained obscure. Using next generation sequencing, we have shown that the *cd* gene in *cd*^1^ carries a long deletion leading to PHS active site destruction. Contrary to the wild type *Canton-S* (*CS*), *cd*^1^ males showed defective long-term memory (LTM) in conditioned courtship suppression paradigm (CCSP) at days 5–29 after eclosion. The number of dopaminergic neurons (DAN) regulating fly locomotor activity showed an age-dependent tendency to decrease in *cd*^1^ relative to *CS*. Thus, in accordance with the concept “from the gene to behavior” proclaimed by S. Benzer, we have shown that the aberrant PHS sequence in *cd*^1^ provokes drastic LTM impairments and DAN alterations.

## 1. Introduction

About half a century ago, Prof. I.P. Lapin discovered the neurotropic effects of kynurenines and suggested the “serotonin-kynurenine hypothesis of depression” [1,2]. Activation of indoleamine 2,3-dioxygenase (IDO), the main enzyme of the kynurenine pathway (KP), decreases the brain serotonin level in favor of kynurenine (KYN) and its products, collectively called kynurenines. It enhances the level of stress and anxiety, being an adaptive reaction aimed at provoking a defensive behavior, but in excess may lead to heavy depression and even to suicide attempts [3]. IDO is induced by pro-inflammatory cytokines during brain infections and neurodegenerative disorders, such as Huntington’s, Parkinsons’s and Alzheimer’s diseases [4,5]. Imbalance of KP products affects neuropathology development by mechanisms that depend on kynurenines’ molecular nature, as well as on changes of their specific ratio within the injured brain area. Diverse effects of kynurenines on nervous activity and behavior were shown for both vertebrates and invertebrates [6].

KP is the major path of tryptophan catabolism. In mammals, it leads to nicotinamide adenine nucleotide (NAD^+^) cofactor synthesis [7]. Insects lack quinolinic acid production and thereby all the subsequent KP stages. This does not affect NAD^+^ level in the body [8]. This is a great advantage for studies on the molecular mechanism of kynurenines’ biological activities while using insect models, such as the *Drosophila melanogaster* KP mutants [9]. For insects, KP is the source of the brown eye pigment xanthommatin (XAN) synthesized from 3-HOK upon its oxidative autodimerization [8,10]. In *Drosophila*, mutations at different KP enzymatic stages affect the eye color (becoming bright red) along with accumulation of specific XAN precursors and lack of products downstream the mutation (Figure 1). *vermilion* (*v*) inactivates the key KP enzyme tryptophan 2,3-dioxygenase (TDO), resulting in 6-fold rise of tryptophan level [11]. In *cinnabar* (*cn*), dysfunction of kynurenine monooxygenase (KMO) leads to 2-fold rise in KYNA and 1.5-fold rise in KYN levels with subsequent lack of 3-HOK [12]. In *cardinal* (*cd*), dysfunction of phenoxazinone synthase (PHS) that governs 3-HOK enzymatic dimerization leads to 2.9-fold increase in 3-HOK level [13,14]. XAN can be produced non-enzymatically due to 3-HOK spontaneous autodimerization [15,16], resulting in the eye color darkening soon after eclosion; Xanthurenic acid (XAA) and cardinalic acid are also synthesized from 3-HOK. The levels of 3-hydroxyanthranilic acid (3-HAA) and anthranilic acid are low in *Drosophila* [8], making *cd* a useful model for specific studies of 3-HOK neurotropic effects.

3-HOK and 3-HAA possess antioxidant activity via inhibiting lipid peroxidation, while the antioxidant power of XAA is significantly less [17,18]. However, in high concentrations, 3-HOK undergoes non-enzymatic oxidative autodimerization, leading to overproduction of reactive oxygen species (ROS), such as hydrogen peroxide inducing apoptosis in neuronal cell culture [19,20]. Total antioxidant activity decreases in *cd*^1^ heads compared to *CS*, while their lipid peroxidation also shows a tendency to decrease with age [16]. This reflects the dual 3-HOK effects on redox processes [21]. KYNA is a non-specific inhibitor of ionotropic glutamate receptors (iGluRs) [22] that ameliorates excitotoxicity caused by glutamate excess. KYNA can form complexes with both mammalian and *Drosophila* iGluR [23], also serving as an α7 nicotinic acetylcholine receptor antagonist [24].

The high 3-HOK/KYNA level in *htt* mutant, a *Drosophila* model for Huntington’s disease, is one of the most crucial factors for neurodegeneration development [25,26]. Rise in 3-HOK/KYNA in flies also leads to age-dependent memory impairments. On days 12–29 of the adult life, *cd*^1^ shows the decrease in learning and 3 h short-term memory (STM) compared to *CS* in the conditioned courtship suppression paradigm (CCSP). On days 21–29, *cd*^1^ demonstrates the development of calyx synaptic pathology. *cn*^1^ does not show the age-dependent STM decay and synaptic pathology, being similar to *CS* [27]. In aged flies, the interpulse interval variance, a marker of neurodegeneration, is the highest for *cd*^1^ and lowest for *cn*^1^. In young flies, similar differences in heat shock-induced brain apoptosis are observed [28]. Thus, 3-HOK accumulation in flies has a general neurotoxic effect progressing with age.

The naive (without courtship experience) males of *CS*, *cn*^1^ and *cd*^1^ have high courtship indices (CI) up to day 29, indicating that their responses to pheromones and visual cues from a fertilized female stay unchanged [27]. At the same time, spontaneous locomotor activity significantly differs in the middle-aged flies. On days 13–29, running speed and run bout time decrease while run frequency increases for *cn*^1^ compared to *CS;* for *cd*^1^ the changes are the opposite. On day 40, the *cd*^1^ index of activity specifically decreases, possibly due to age-dependent neurotoxicity development [29]. Thus, it is crucial to differentiate KYNA and 3-HOK effects on spontaneous locomotion activity, courtship activity, memory processes and neurodegeneration.

Although *cd*^1^ mutation has been known about since the birth of *Drosophila* genetics [30] and is used as a marker gene in many strains, its molecular nature remains unknown. PHS activity associated with vesicular structures in the pigment cells called the pigment granules specifically rises at the pupal stage when XAN synthesis occurs. This activity decreases both in *cd* and a number of other XAN-deficient strains, such as *v* and *cn* mutants, possibly due to the lack of 3-HOK that may activate PHS [31]. According to other researchers, 3-HOK non-enzymatic dimerization decreases in head homogenates of the *cd* strain, while 3-HOK enzymatic dimerization does not change compared to the wild type [32]. However, they estimated the “PHS activity” in a soluble fraction that lacked the pigment granules. Hence, it was probably the non-enzymatic 3-HOK dimerization activity. The role of PHS in *Drosophila* 3-HOK dimerization was proved using the transgenic wild type *cd* gene, the expression of which completely restores the ommochrome synthesis in several loss-of-function *cd* mutants [33].

*Drosophila* PHS is predicted to have N-terminal transmembrane domain and heme peroxidase domain. Being localized in pigment cells, namely in type I pigment granules, PHS prevents Ago2 loading to protein-siRNA complex. siRNA-mediated silencing is required for normal vesicle trafficking and ommochrome synthesis. At the same time, mutations in the *v*, *cn* and *cd* genes specifically inhibit this silencing pathway in the fly eyes [33]. This indicates positive feedback between the ommochrome level and pigment granule trafficking. Surprisingly, the level of protein-bound 3-HOK (PB-3-HOK) in the *cd*^1^ brains and head capsules appears to be several times lower compared to *CS,* though it remains higher compared to *v*^1^ and *cn*^1^ [29]. Hence, *Drosophila* PHS may also participate in 3-HOK conjugation to some proteins.

To reveal the molecular nature of *cd*^1^ mutation, we performed the next-generation sequencing (NGS) of the *cardinal* gene (*cd*) for *CS* and *cd*^1^ strains. We also studied the relative abilities of *CS*, *cn*^1^ and *cd*^1^ to learn and form LTM in CCSP. The age-dependent changes in learning and memory abilities were estimated for flies trained on days 5, 13, 21 and 29 after eclosion. The number of tyrosine hydroxylase (TH)-positive dopaminergic neurons (DAN) within the brain areas regulating courtship memory and locomotor activity was counted in young and aged *CS* and *cd*^1^.

## 2. Results

### 2.1. PCR Mapping of the cd Gene

To reveal possible *cd* allelic variants in *Drosophila* populations, PCR mapping of the *cd* gene was performed for two different single males of *CS* and *cd*^1^ strains (Figure 2). All bands were present in *CS*, their size being approximately equal to the theoretically expected value (see Appendix A). Band 3 was weak in *CS* #2, which might reflect some primer non-specificity. Similar bands were observed in *cd*^1^. However, band 5 was nearly absent in *cd*^1^ #2. There was a smeared band below, which could also be seen in *cd*^1^ #1. Hence, the *cd* gene in *cd*^1^ seems to carry some polymorphism, probably a deletion, within area 5.

The presence of deletion was confirmed for the *cd*^1^ amplicons, including the whole *cd* gene with the flanking sequences (Appendix A). The amplicon I (1f–7r) in *CS* was compared to the amplicon I’ in *cd*^1^ (8f–7r) with nearly matching borders and was found to be longer, and the band I’ was double. Surprisingly, there were no visible interstrain differences for the amplicon II (2f–7r) lacking *cd* 5’ end, probably corresponding to a different copy of the *cd* gene. Thus, *cd*^1^ seems to carry at least two inactive copies of *cd*, one of which lacks a significant part of the gene.

### 2.2. The cd Gene Sequencing in CS and cd^1^

The results of sequencing of four amplicons are presented in Figure 3. For the wild type strain, only one amplicon (I) covering the whole gene with flanking sequences was assayed. Compared to the reference genome sequence, *CS* carried ~50 single nucleotide polymorphisms (SNPs) within the *cd* gene and flanking 5′ and 3′ areas (Appendix A). Most of them were synonymous, some were upstream gene variant modifiers, being localized within introns and the intergenic spacer, and four were missense mutations leading to amino acid substitutions. All of them were beyond the predicted PHS active site and presumably could not affect its function. There were also two (19 bp and 2 bp) A/T-rich indels downstream *cd*. Sequencing depth (DP) was similar for all positions. C(1504)/T SNP within the second part of the *CS* 3r primer binding site probably caused the non-optimal amplification of band 3 in this strain.

The picture was significantly different for *cd*^1^. Here, the amplicon I’ (8f–7r), the borders of which nearly matched those for amplicon I (1f–7r), was analyzed. There was a drastic drop of DP beginning from 1944 bp till the reverse primer binding site, being the evidence of the large deletion, though some areas showed local increase in DP starting from ~200 bp after the beginning of the deletion. The non-zero DP of the *cd* 3’ area corresponded to the double band I’ with the faint heavy part of the band. Thus, it was hard to localize the 3’ end of deletion for this amplicon. The sequence of amplicon II was nearly the same as for *CS*, except that the *cd* 5 end was not amplified in II, and SNP distribution in I (*CS*) and II (*cd*^1^) significantly differed from that of I’ (*cd*^1^). There was no evidence of a long deletion such as in I’: DP was similar for all positions. This confirms that II is a part of the second inactive *cd* copy in *cd*^1^.

To identify the 3’ end of the deletion within the *cd* gene in *cd*^1^, we performed additional assay for amplicon III (8f–8r) that was obtained independently for the other fly samples and was significantly longer than I and I’. Again, we observed a significant drop of DP from 1944 bp, up to 2080 bp (including the AGGTGG... TTCAAT area; see Text S2). The deletion was flanked by GG/CC repeats. This makes possible formation of a loop with a stem including A/T pair of the deleted sequence. The deletion was located within band 5 (1716–2279 bp). On PCR in *cd*^1^, the light smeared part of the double band 5 was several hundred bp shorter compared to the predicted size (see Figure 2). T(1726)/G SNP was within 5f primer binding site in *cd*^1^; hence the double band also might be an effect of non-specific amplification.

The found deletion covered 137 bp of the *cd* gene, including the key residues of the active site (Figure 4), such as Arg 333/542, Asn 420/629, Arg 423/632 (heme fixation) and His 336/545 (interaction with the heme group, participation in catalysis). The residue numbers are given according to their order in the PHS model structure/full protein sequence; see also [16] for the description of PHS active site and catalytic mechanism. Moreover, the deletion induced a frameshift and synthesis of the abnormal 10 AA peptide (AILALGGAWH) followed by translation termination (Appendix A). As a result, the mutant PHS should lack a C-terminal part including most residues of the active site. Both I’ and III amplicons also had T(20) > G substitution within the binding site of the transcription factor Adf-1, which might affect *cd* expression in *cd*^1^.

In summary, *cd*^1^ carries at least two non-functional *cd* copies—*cd*(*A*) with T(20) > G change in the promoter area and the deletion disrupting the enzyme catalytic site, and *cd*(*B*), for which the molecular nature of impairment remains unknown.

### 2.3. The cd Gene Expression in Late Pupae

To reveal whether *cd*^1^ mutation impairs *cd* mRNA synthesis, we performed a semi-quantitative RT-PCR analysis of *cd* expression in both strains at the late pupal stage, when PHS is actively involved in XAN production. We did not observe a decrease in *cd* activity in *cd*^1^ compared to *CS*, both for the total RNA and poly(A) form (Figure 5). Thus, either T(20) > G substitution in *cd*^1^ does not impair *cd*(*A*) synthesis and maturation, or *cd*(*A*) mRNA decrease is compensated by *cd*(*B*) transcription activity. 

### 2.4. Learning and Long-Term Memory Formation

The learning acquisition and long-term memory (LTM) retention were assessed for *Drosophila* males in CCSP at 5, 13, 21 and 29 days of adult life. On day 5, both the wild type strain *CS* and the mutant strain *cd*^1^ showed courtship index (CI) decrease just after training (0D). The difference between CI of naive and trained flies gradually becomes insignificant with age and time after learning, indicating memory decay (Appendix A). In *CS*, the learning index (LI) calculated at 0D point (LI(0D)) remained significantly different from zero up to 21 days after eclosion (Figure 6). On day 21, LI(2D) and LI(8D) did not differ from zero, and LI(8D) was significantly lower than LI(0D). The above indicated 2D and 8D LTM were impaired in middle-aged *CS*. In *cd*^1^, both 2D and 8D LTM were absent during the life span, along with the intact learning ability. On day 5, *cd*^1^ showed LI(8D) decrease compared to *CS*. On day 29, LI did not statistically differ from zero for both strains and time points, except for *cd*^1^ LI(0D), indicating preservation of its ability to learn while training for 5 h. In the other KP mutant *cn*^1^, both learning ability and 2D LTM were preserved up to 21 days, while 8D LTM was impaired during the whole studied life period (Appendix A). Thus, both KP mutants showed 8D LTM impairments, while *cd*^1^ also lacked 2D LTM starting from a young age. 

### 2.5. Dopaminergic Neurons in Adult Drosophila Brain

A 24 h LTM formation in CCSP paradigm is known to depend on aSP13 DAN, innervating the tip of the mushroom body (MB) *fru^+^* γ-lobe [34]. aSP13 neurons belong to the protocerebral anterior medial (PAM) cluster. Among the other DAN innervating MB are the protocerebral posterior lateral (PPL) and medial (PPM) clusters [35]. The *scarlet* mutant with a defect in 3-HOK transport to pigment cells is characterized by ROS increase, decreased climbing index and age-dependent loss of DAN belonging to a PPL1 cluster [36]. It is reasonable to assume that 3-HOK accumulation in *cd*^1^ leads to age-dependent PAM degeneration, possibly causing LTM impairment. To check this hypothesis, we counted the cell number in several DAN clusters for *CS* and *cd*^1^ on days 5 and 29.

The distribution of tyrosine hydroxylase (TH)-positive DAN clusters in the 5-day-old and 29-day-old *Drosophila* brains is shown in Figure 7. As expected, the cell bodies of the TH-positive neurons were located above the antennal lobes (AL), near the superior medial protocerebrum (SMP). Their processes innervated the horizontal lobes of the MB, especially the medial tip of the β′ lobe (β′L), like in [35], where the processes extending from both hemispheres join together. The semicircular TH-positive tracts that surrounded SMP were less visible for GFP-positive cells, whereas the tracts connecting PAM cells to β′L were seen better (Appendix A). PPL1 and PPM2 neurons were located lateral and medial to calices (Cal), whereas PPM3 neurons were located somewhat below Cal. Cell bodies formed clusters tightly connected in groups by their processes. In each hemisphere, the PAM cluster was subdivided into two approximately equal branches (see 3D reconstruction in Appendix A).

The average numbers of TH-positive cells on days 5 and 29 of the *Drosophila* adult life were somewhat lower compared to previous data [35], probably due to lower resolution in our case. Mao and Davis gave the approximate values of 100, 12, 8 and 6–8 for PAM, PPL1, PPM2 and PPM3 neurons, respectively, whereas in our study these clusters in the young *CS* included approximately 43, 10, 6 and 5 neurons (Figure 8). Other DAN clusters were not considered. Proper orientation of the brains was reported to be crucial for reproducible quantification of DAN [37]. For the Gal4 > UAS strain expressing GFP in DAN and serotoninergic neurons (SRNs), the number of cells in PAM clusters was larger: 74 on average and up to 100. SRNs might have only a minor impact on this number [38,39]. We did not observe age or strain-specific differences in the average number of PAM cells responsible for courtship memory, as well as in PPM3 cells innervating the central complex (CC). The average number of PPM2 cells was slightly increased in 5-day-old *cd*^1^ relative to *CS*. On day 29, the number of PAM cells was non-significantly decreased in *cd*^1^ relative to the wild type strain, while the decrease of PPL1 cell number was nearly significant (*p* = 0.057). The observed tendency probably reflected the age-dependent DAN degradation in *cd*^1^ due to oxidative stress.

## 3. Discussion

The gene-to-behavior approach suggested by S. Benzer in 1971 raises the question how the one-dimensional information contained in genes is translated into a multi-dimensional spatial structure of the central nervous system realized in complex behavioral patterns [40]. The *Drosophila* KP mutants serve as an excellent model to investigate the causal chain connecting gene structure changes to physiological and behavioral alterations. Among them is the *cd*^1^ mutation. *cd*^1^ has not been sequenced up to date, despite being actively used as a marker gene and in HD modeling.

Our NGS data on the *cd*^1^ *cd* structure revealed the complex nature of this mutation. There are at least two different *cd* copies in the genome of the *cd*^1^ flies, both inactive or only partially active. One mutant copy, *cd*(*A*), carries a deletion disrupting PHS active site and an SNP within the predicted Adf-1 binding site. Another copy, *cd*(*B*), is similar to the wild type *cd*, except for its 5’ end that has not been sequenced. As amplicon I contains a 5’ end and lacks the middle part of amplicon II, we suppose that *cd*(*B*) may carry some alterations in the 5’ area. The double bands for 1f–7r and 8f–7r amplicons in *cd*^1^ may also indicate some polymorphisms within the 1f/8f binding area. We did not observe different eye color phenotypes in the *cd*^1^ population; hence, both *cd*(*A*) and *cd*(*B*) are recessive and do not recombine to produce the wild type *cd* form. As *cd*^1^ was cantonized before experiments and SNP distribution is very similar in *cd*(*B*) and *CS*, *cd*(*B*) may be the non-functional part of the wild type *cd* gene obtained in the process of cantonization, 

There can be two different ways in which both *cd* copies are arranged in the genome of *cd*^1^ (Figure 9). In the first case (Figure 9a), they are located in one locus and can be redistributed in the population in different combinations. In the second case (Figure 9b), two different *cd* copies are located in different parts of the genome, within the 3R:94A1-94E2 cytogenetic region (Flybase data). For single males, we observed differences in the brightness of band 5, which might indicate the presence or absence of short (*A*) and long (*B*) *cd* forms (see Figure 2). Both II and III *cd* amplicons are followed by the *Cyp6d4* gene, indicating their equal position in the genome. These facts argue in favor of the first case (Figure 9a). Alternatively, there may be a long insertion in the *cd*^1^ genome carrying both *cd* and *Cyp6d4*, where *cd* is represented by *A* or *B* forms. In our further research, we plan to perform in situ hybridization to visualize the deleted fragment of the *cd* gene on the *cd*^1^ polytene chromosomes to check whether it resides only in half of chromatids at *cd* locus (case 1), or in all chromatids but at some different loci (case 2). Isolation of different stocks and sequencing is also possible in the future.

The deletion of the 3′ end covers a significant part of PHS active site, probably leading to a drastic decrease in PHS catalytic activity and 3-HOK accumulation. Both facts are in agreement with experimental data [13,14,31]. Here, we have shown that the level of *cd* RNA at the late pupal stage was not decreased in *cd*^1^ compared to *CS*. PHS may also preserve some functional activity, causing the *cd*^1^ eye darkening soon after eclosion—the effect usually explained by 3-HOK autodimerization.

The prolonged action of 3-HOK induces the oxidative stress and neuronal apoptosis [19,20]. Seemingly, this causes functional disturbances in *cd*^1^ manifestations, such as courtship song defects [28], and possibly STM disturbance in aged flies [27]. Heat shock induces apoptosis in young *cd*^1^ flies [28]. In addition, it impairs STM in *cd*^1^ [41]. In contrast, *cn*^1^ shows the neuroprotective and anti-apoptotic activity [28]. In our study, *cn*^1^ demonstrated normal 2D STM in CCSP up to day 21 of adult life, but both KP mutants had impaired 8D LTM. The lack of 8D LTM was also shown for 5-day-old *v*^1^ with absence of kynurenines and intact 3-h STM [42].

The mechanisms of CCSP are complex and are still not fully understood [43]. Figure 10 gives a simplified scheme of the brain areas participating in courtship behavior [34,44,45]. Let us highlight several key elements crucial for CCSP memory formation:

The olfactory sensory neurons Or67d for anti-aphrodisiac cis-vaccenyl acetate (cVA), which is crucial for courtship suppression [34].The visual projection neurons (VPN) from the optic lobe (OL).MB calyx (Cal) and intrinsic neurons. MB are responsible for the classical olfactory conditioning [46] and are not required for leaning in CCSP [47,48]. However, Kenyon cells (KCs) of MB receive inputs not only from the olfactory AL. Subesophageal ganglion (SEG) responsible for taste sensitivity activates the dorsal accessory calyx (dAC) [49]. γd neurons of the ventral accessory calyx (vAC) participate in CCSP STM formation, receiving visual inputs from OL [48,50]. These neurons project to the MB γ5 area, crucial for courtship suppression [51].The intrinsic neurons of MB projections to the MB output neurons (MBONs). Their synaptic contacts are tightly regulated by DANs, activated by unconditioned stimuli, such as electric shock or sugar. They do not encode specific modality of a stimulus, rather its positive or negative valence. After learning, the mutual MBON effect is shifted towards attraction or avoidance, biasing the fly’s behavior [52].MBON project to the higher brain integrative centers, such as SMP, lateral protocerebrum (LP) and lateral horn (LH), also containing about 90% of the DANs dendritic arbors [50,52]. LP is the major site for the integration of discrete sensory stimulus, aggregated by P1 cluster, which triggers male courtship [44,45].

In our study, we used a CCSP retraining protocol, where a mated female served a trainer [53,54]. According to Keleman and coauthors, courtship learning in this case occurs as an enhanced response of a male after unsuccessful courtship to cVA released from a female. The active rejection by a female is necessary for training, while cVA is required for memory performance. aSP13 DANs of a PAM cluster play a key role in CCSP. They innervate the tip of the MB γ-lobe (γ5 area), expressing dopamine receptors DopR1 [34]. The other name of aSP13 is PAM-γ5 [51]. aSP13 prolong potentiation of MBγ–M6 MBON synapses, being the physical basis of the courtship STM [55]. The prolonged aSP13 activation leads to protein-synthesis-dependent LTM consolidation, dependent on the Orb2 protein [56,57].

Additional MB structures may participate in memory retrieval. Blocking the output of γ neurons disrupts STM, leaving LTM intact, while some other MB neurons, such as the neurons of αβ lobes, probably govern LTM expression [54]. Sleep is the other factor crucial for memory consolidation in CCSP. aSP13 are activated after courtship by a specific class of sleep-promoting vFB neurons in the fan-shaped body of the central complex (CC). An activation of vFB neurons at a specific time interval after learning is sufficient for LTM consolidation [58].

Our data also suggest different mechanisms for the *Drosophila* STM and LTM. In young flies, 8D LTM was absent in both KP mutants and *v*^1^, whereas 2D LTM was intact in *v*^1^ and *cn*^1^, and 3-h STM was intact in all the studied strains [27,42]. LTM preservation up to day 8 seems to require robust memory mechanisms impaired to varying degrees in KP mutants. Although KYNA accumulation in *cn*^1^ has neuroprotective effects, it also damages LTM storage or retrieval 8 days after learning, probably due to KYNA inhibitory action on iGluR or nicotinic receptors. M6-MBONs are putatively glutamatergic neurons [51], so KYNA excess can inhibit the output from the γ5 area and/or its translocation to other MB regions, crucial for LTM expression. *cd*^1^ with accumulation of the neurotoxic 3-HOK shows the most pronounced memory defects. The lack of 8D memory in *cd*^1^ may reflect the faster decay of its LTM compared to *v*^1^ and *cn*^1^. The interstrain differences were seen on days 5 and 13. At the same time, STM decrease and calyx neuropathology was observed only in aged *cd*^1^ [27]. Hence, LTM impairments in *cd*^1^ are ahead of the visible brain damage development and possibly occur at a more subtle neuronal level. As STM is also impaired in 5-day-old *cd*^1^ after heat shock, 3HOK-dependent neurotoxicity seems to occur early enough, being manifested under special conditions.

DAN clusters that regulate CCSP memory formation and consolidation may be selectively susceptible to the toxic effects of 3-HOK. However, we did not see any difference in PAM number for young *CS* and *cd*^1^. A slight increase in PPM2 number was observed in young *cd*^1^. These neurons innervate the part of the medial protocerebrum and SEG [35]. In aged flies, there was a trend of PAM decrease in *cd*^1^ relative to *CS*. However, the difference was non-significant, and LTM was equally impaired in both strains. As the number of PAM neurons did not change in *CS* upon aging, the LTM impairments on day 29 seem not to be due to death of PAM neurons. However, we cannot rule out that LTM lack may be caused by the decrease of PAM activity. The total number of TH-positive DAN did not change in several *Drosophila* models for PD, though their functional activity decreased in aged flies [37]. At the same time, both the loss of PPL1 neurons and locomotor defects were shown for *scarlet* mutant, a PD model with 3-HOK accumulation and increased ROS level [36]. In 29-day-old *cd*^1^, we also observed a trend of PPL1 number decrease relative to *CS*. This trend may become even stronger in 40-day-old *cd*^1^ with a decrease in spontaneous locomotor activity [29]. While PAM is needed for courtship memory, PPL1 is unlikely to be involved in it [48]. 

OL is connected to the γ5 area responsible for courtship memory [48,50]. Hence, visual signals from a female may be important for the courtship suppressions. Environmental light is also critical for LTM, which is impaired when flies are constantly kept in darkness after training. Memory consolidation is regulated by the light-driven activity of the transcription factor CrebB in αβ and γ lobes of MB. CrebB activity in αβ lobes is also required for LTM maintenance [59].

All the studied KP mutants have impaired synthesis of brown eye pigment XAN and its derivatives. In *cd*^1^, XAN is synthesized soon after eclosion, either due to 3-HOK non-enzymatic autodimerization, or due to PHS residual activity. All KP mutants showed a decrease in the PB-3-HOK form, both in their brains and head capsules [29]. The physiological role of such a 3-HOK form is not clear, but it may have some protective or regulatory functions. Since courtship activity is the same for KP mutants and does not change with age, the visual system regulating male orientation toward females seems to be unaffected [27]. However, some minor disturbances of visual processing due to pigmentation defects or 3-HOK-dependent neurotoxicity may be crucial for LTM formation and preservation.

In summary, the deletion of the part of the *cd* gene in *cd*^1^ encoding the PHS catalytic site explains the three-fold increase in free 3-HOK that was previously shown for this mutant. Some residual PHS activity may preserve being possibly associated with the other *cd* copy. Checking this assumption needs further research, as well as mapping different *cd* copies in the *cd*^1^ genome. It also remains to be found whether PHS plays some role in 3-HOK conjugation to proteins. Studying the role of DAN in 3-HOK-induced memory disturbances requires specific suppression of DAN cluster activity in *cd*^1^. The other task for the future is checking the influence of visual capacity on the abilities of KP mutants to form and preserve LTM. Disclosing mechanisms of behavioral impairments for the *Drosophila* KP mutants serving as model objects for different neural disturbances helps to uncover the molecular mechanisms of neurodegeneration and cognitive disorders.

## 4. Materials and Methods

### 4.1. Drosophila Strains

Fly strains were taken from Biocollection of the Pavlov Institute of Physiology Russian Academy of Sciences, Saint Petersburg, Russia. The following strains were used:*Canton-S* (*CS*)*,* the wild type strain; dark red eyes.*cinnabar* (*cn*^1^)—this carries a mutation in the kynurenine monooxygenase gene; lack of 3-HOK, KYNA excess, bright red eyes.*cardinal* (*cd*^1^; Bloomington *Drosophila* Stock Center, #3052)—this carries a mutation in the *cd* gene; increase in 3-HOK level, bright red eyes after eclosion; in 3-day-old flies the eye color becomes dark red.

*cn*^1^ and *cd*^1^ strains were out-crossed to *CS* for nine generations. All strains were raised on a standard yeast–raisin medium with 8 a.m.–8 p.m. daily illumination at 25 ± 0.5 °C.
4.Strains from Bloomington Drosophila Stock Center:

#7009: w [1118]; P{w[+mC] = Ddc-GAL4.L}Lmpt[4.36]. GAL4 is expressed in dopaminergic (DAN) and serotoninergic (SRN) neurons.

#32186: w[*]; P{y[+t7.7] w[+mC] = 10XUAS-IVS-mCD8::GFP}attP40. This contains GFP, the expression of which is driven by GAL4.

### 4.2. DNA Extraction

DNA was extracted from 10 females (for sequencing; 300 µL buffer) or single males (for PCR mapping; 200 µL buffer). Flies were homogenized in DNA extraction buffer (0.1 M Tris, 0.1 M NaCl, 0,05 M EDTA (pH 8.0), 0.2 M sucrose, 0.5% SDS, 0.5% diethyl pyrocarbonate) and heated for 30 min at 65 °C. After addition of sodium acetate (pH 5.2, the final concentration 1.8 M), homogenates were incubated at 0 °C for 30 min and centrifuged at 14,000× *g* (MPW-65R microcentrifuge, Warsaw, Poland). An equal volume of 96% ethanol was added to supernatant. DNA was precipitated by centrifugation at 14,000× *g* for 5 min and washed with 70% ethanol. DNA concentration was measured at 260 nm using spectrophotometry (Eppendorf BioPhotometer, Hamburg, Germany).

### 4.3. Polymerase Chain Reaction

According to Flybase (www.flybase.org, accessed on 1 December 2021), *cd* location is 3R:22,694,959…22,697,990 [+]. Upstream *cd*, *Nup133* gene is located (3R:22,690,607..22,694,772 [+]) encoding a Nup133-like nuclear pore complex protein. Downstream *cd*, *Cyp6d4* gene is located (3R:22,698,319..22,700,307 [+]) encoding a heme binding protein that belongs to the cytochrome P450 family. To cover the *cd* gene with flanking 5′ and 3′ areas, a series of gene-specific primers was designed using NCBI Primer Blast tool and *D. melanogaster* genomic reference sequence. For all primers, the predicted melting temperatures are within 59–61 °C. DNA was amplified with Long PCR Enzyme Mix (ThermoFisher Scientific, Waltham, MA USA, #K0182) according to the manufacturer’s guidelines using Verity 96-Well Thermal Cycler (Applied Biosystems, Waltham, MA USA). Primer sequences, positions of overlapping amplicons and PCR parameters are given in Appendix A.

### 4.4. DNA Purification and Sequencing

The PCR products were separated by agarose gel electrophoresis using Agagel Mini system (Biometra, Göttingen, Germany). The products of interest were extracted from gel using QIAquick Gel extraction kit 250 (QIAGEN, Cat. No 28706). DNA concentration before sequencing was 7–20 µg/mL. DNA amplicons were purified using AMPure XP beads (Beckman Coulter, Indianapolis, IN, USA) and fragmented using dsDNA Fragmentase (NEB, Ipswich, MA USA). The fragmentation reaction was carried out in 10 µL volume for 10 min at 37 °C using 25 ng of purified DNA. The fragment-DNA libraries were prepared using TruSeq Nano DNA LT Library Prep Kit (Illumina, San Diego, CA USA), following the manufacturer’s protocol. The prepared libraries were sequenced on Illumina HiSeq 2500 system the paired-end 2x130 mode. The raw sequence data were deposited in NCBI, Sequence Read Archive, ID: PRJNA840853.

### 4.5. Bioinformatics Analysis

Assembly of sequences and annotation were performed using *D. melanogaster* genome reference (dm6 genome assembly; NCBI GenBank). FastQC [60] was used to check the quality of reads. Low-quality reads were deleted with the Trimmomatic tool [61] using the following filtration parameters: leading: 15; trailing: 15; sliding window: 4:22; minlen: 36 or 40. Sequence assembly was performed for paired reads with the help of the BWA-MEM algorithm [62] and samtools/bcftools utilities [63]. IGV software [64] was used for visualization of the sequencing data. Only the area corresponding to the amplicon position was analyzed, neglecting 5′ and 3′ regions with low sequence depth (DP). Substitutions with DP < 15 or quality < 33 (Phred33 quality score) within the amplicon area were omitted from the analysis. The effects of single nucleotide polymorphisms (SNPs) and indels were predicted with the help of SnpEff [65].

Search of transcription factor binding sites was performed with the help of PROMO 3.0 software [66]. Homology modeling of mutant PHS 3D structure was performed with MODELLER 9.14 software [67] using goat lactoperoxidase as a template (PDB ID: 2ojv). The modeling procedure of the wild type *Drosophila* PHS is described in detail in [16].

### 4.6. RNA Extraction, Reverse Transcription and Semi-Quantitative Real-Time Polymerase Chain Reaction (sq-RT-PCR)

RNA was extracted from late pupae (before eclosion). Four male pupae were homogenized in 300 µL of TRI reagent (MRC, TR 118). Total RNA was treated with DNAse I in columns and purified using Direct-zol RNA MiniPrep (Zymo Research, R2050), in accordance with the manufacturer’s instructions. A total of 1 µg of RNA was used for cDNA synthesis with the help of MMLV reverse transcriptase (Evrogen, SK021). To assess the level of total RNA and the polyadenylated RNA form, we used random hexamer and oligo(dT) primers, respectively. Three biological replicates were made in each case. sq-RT-PCR was performed with the help of qPCR premix (Evrogen, PK156L), using StepOnePlus Real-Time PCR system (Applied Biosystems). cDNA for the *Drosophila* ribosomal protein rpl32 served as an endogenous control. The *cd* PCR amplicon spans the junction between exons 2 and 3. Primer sequences and PCR parameters are given in Text S3.

### 4.7. Long-Term Memory in Conditioned Courtship Suppression Paradigm

Learning acquisition and LTM retention was estimated in CCSP [53,68] modified for LTM [54]. For each strain, all males were divided into several independent groups: 1. Naive (without courtship experience); 2. 0D (learning, immediately after training with females); 3. 2D LTM (2 days after training); 4. 8D LTM (8 days after training). Each group consisted of four independent subgroups for training experiments at the age of 5, 13, 21 and 29 days after eclosion. Each subgroup contained about 20 flies. Males were kept separately in vials with standard food medium until the day of experiment. Fertilized 5-day-old *CS* females served as objects of courtship.

For training, a naive male and a female were placed together into a food-containing vial for 5 h. The free space of movement was about 20 mm (diameter) × 10 mm (height). After training, a male was either tested immediately with a new female (learning) or kept separately in a food-containing vial for 2 days (2D LTM) or 8 days (8D LTM). During test, a male and a female were placed together into a Perspex chamber (diameter 15 mm, height 5 mm). After 45 s of adaptation, the ethogram of a male courtship behavior was recorded for 300 s, manually fixing specific courtship elements (orientation, vibration, attempted copulation), as well as of elements unrelated to courtship (running, preening, rest). A specifically designed *Drosophila* courtship Lite (DCL) software was used to decipher and analyze the recorded data (Kamyshev, 2006). The program is freely available from the author (nkamster@gmail.com) upon request.

To quantitatively estimate the learning acquisition and memory formation, courtship index (CI) was calculated for each male as a percentage of test time spent in courtship. Learning index (LI) was calculated using the formula:LI = [(CI_N_ − CI_T_)/CI_T_] × 100% = (1 − CI_N_/CI_T_) × 100%
where CI_N_ is the average CI for naive males, and CI_T_ is the average CI for males after training. The naive and trained males were the same age.

Statistical analysis was performed using two-sided randomization test at significance level α = 0.05 [69] using DCL software. The following criteria of memory preservation were used: 1. LI was not significantly decreased with time compared to LI just after training. 2. LI was statistically different from zero. 3. LI for a mutant strain was not significantly decreased compared to the corresponding LI for the wild type strain *CS*.

### 4.8. Dopaminergic Neuron Visualization

The brains of 5-day-old and 29-day-old adult *Drosophila* males were isolated in PBS buffer (pH 7.5) using fine needle-sharp style #5 tweezers (Merck, T4412) and fixed in 4% PFA on PBS for 1 h at RT, according to [70], without the freezing step before staining. The following antibodies were used: primary: rabbit anti-*Drosophila* tyrosine hydroxylase (TH) antibody (Abcam, ab128249; dilution 1:200), mouse anti-*Drosophila* cysteine string protein (CSP) antibody (courtesy of E. Buchner, Germany; dilution 1:20); secondary: goat anti-rabbit Alexa Fluor 633 (Invitrogen, A21071; 1:200), goat anti-mouse Alexa Fluor 488 (Invitrogen, A32723; 1:200). Incubation was performed at 4 °C on a shaker for 3 days with primary antibodies and for 2 days with secondary antibodies. The long incubation period was used to improve the brain staining, similar to that in [29]. Cell nuclei were stained with DAPI (1.2 µg/mL PBS). The brains were mounted using Vectashield mounting medium (Vector laboratories) and scanned frontally by the confocal laser scanning microscope (LSM 710 Carl Zeiss; Confocal microscopy Resource Center; Pavlov Institute of Physiology Russian Academy of Sciences, Saint Petersburg, Russia). Scanning was performed using X20 objective (6 µm Z step) and X63 objective (2 µm Z step). Confocal images were analyzed using Fiji software. The number of TH-positive neurons was counted using Cell Counter plugin. TH-positive cells belonging to different DAN clusters were visually identified based on [35]. The average number of cells for each TH-positive cluster was counted for each brain hemisphere (or the number for one hemisphere, if the other was damaged). Statistical analysis was performed using two-sided t-test (*p* < 0.05).

## Figures and Tables

**Figure 1 ijms-23-12356-f001:**
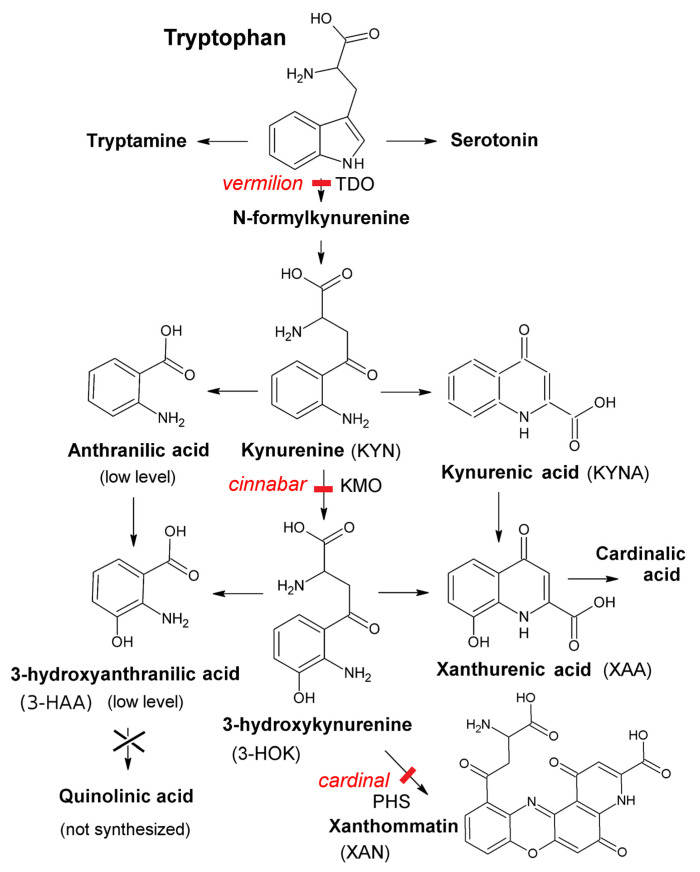
Kynurenine pathway in *Drosophila*.

**Figure 2 ijms-23-12356-f002:**
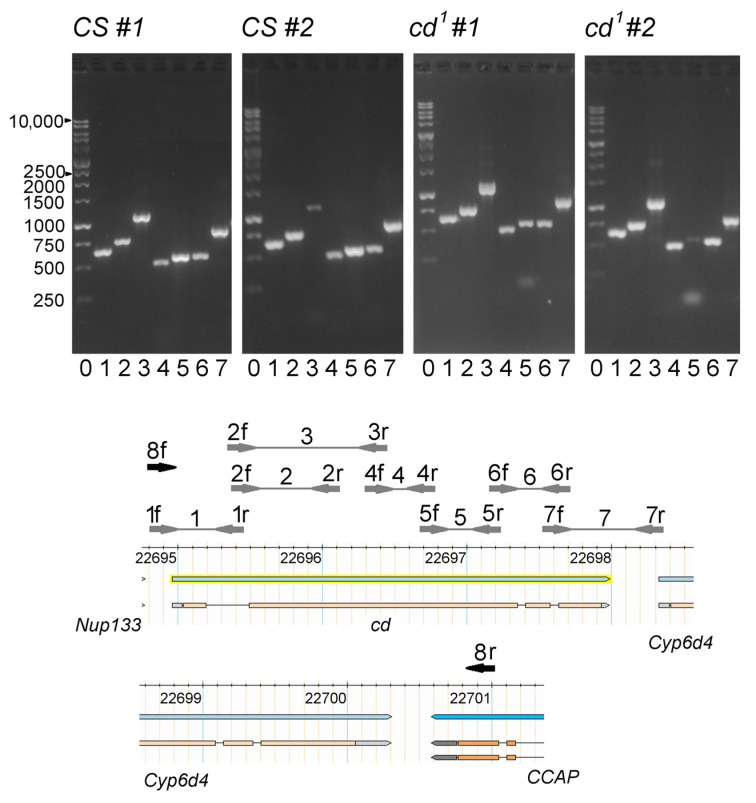
PCR mapping of the *cd* gene in *CS* and *cd*^1^ mutant. 0: 1 kb ladder. 1–7: the primer pairs (f—forward, r—reverse). Positions on the X chromosome are shown in kb (FlyBase data). 8fr—the amplicon III (see the text).

**Figure 3 ijms-23-12356-f003:**
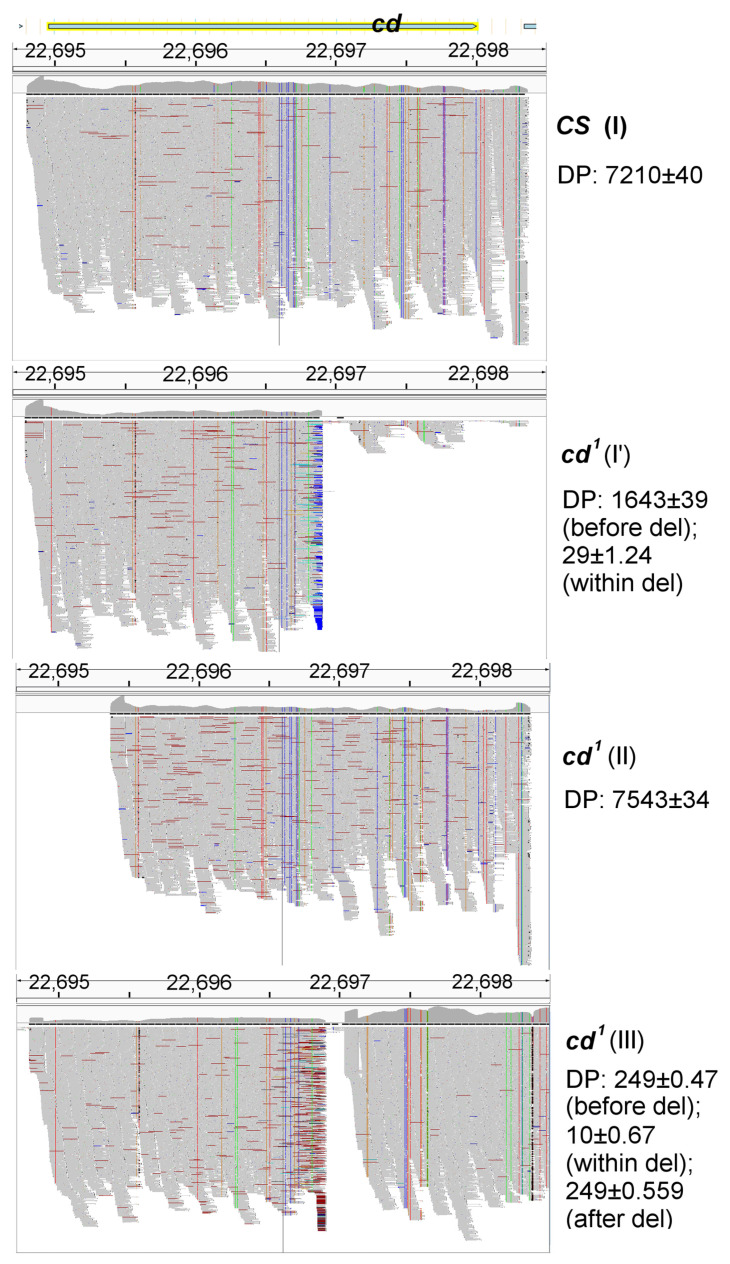
Aligned reads of the *cd* gene amplicons. Reads are shown in grey. Single nucleotide polymorphisms (SNPs) and misaligned reads are shown in other colors. DP—sequencing depth, del—deletion. Positions on the X chromosome are shown in kb. The amplicons are indicated by Roman numerals.

**Figure 4 ijms-23-12356-f004:**
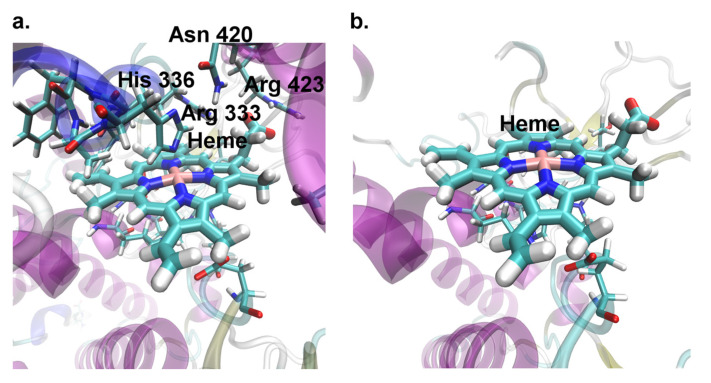
Three-dimensional model structure of the *Drosophila* phenoxazinone synthase (PHS) active site. (**a**). *CS*; (**b**). *cd*^1^ (amplicon III).

**Figure 5 ijms-23-12356-f005:**
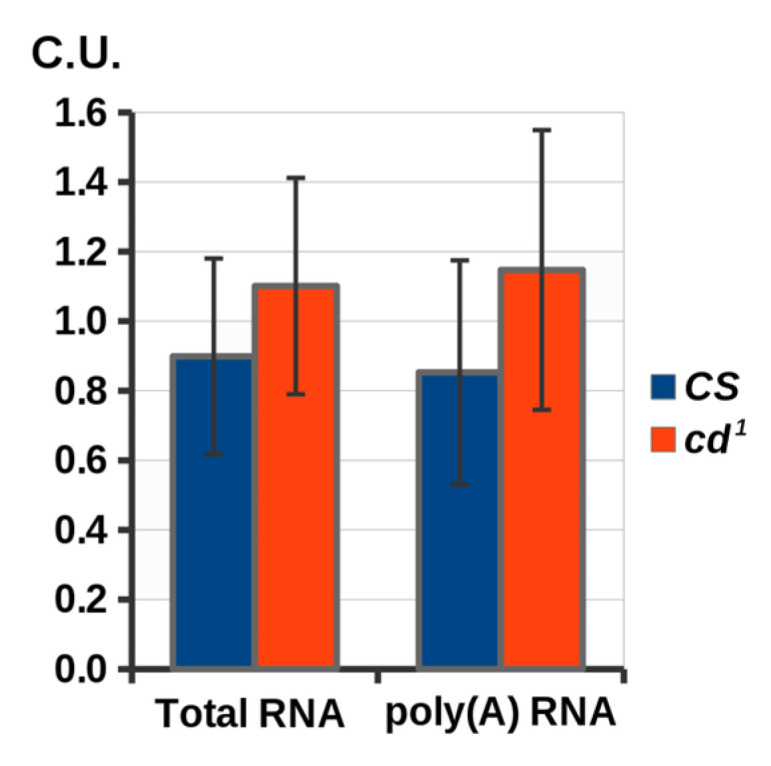
RT-PCR analysis of *cd* expression in *CS* and *cd*^1^ late pupae. *Y* axis: expression level (in conventional units). Standard error is shown. No statistical differences were observed (two-sided *t*-test; *p* < 0.05, n = 3).

**Figure 6 ijms-23-12356-f006:**
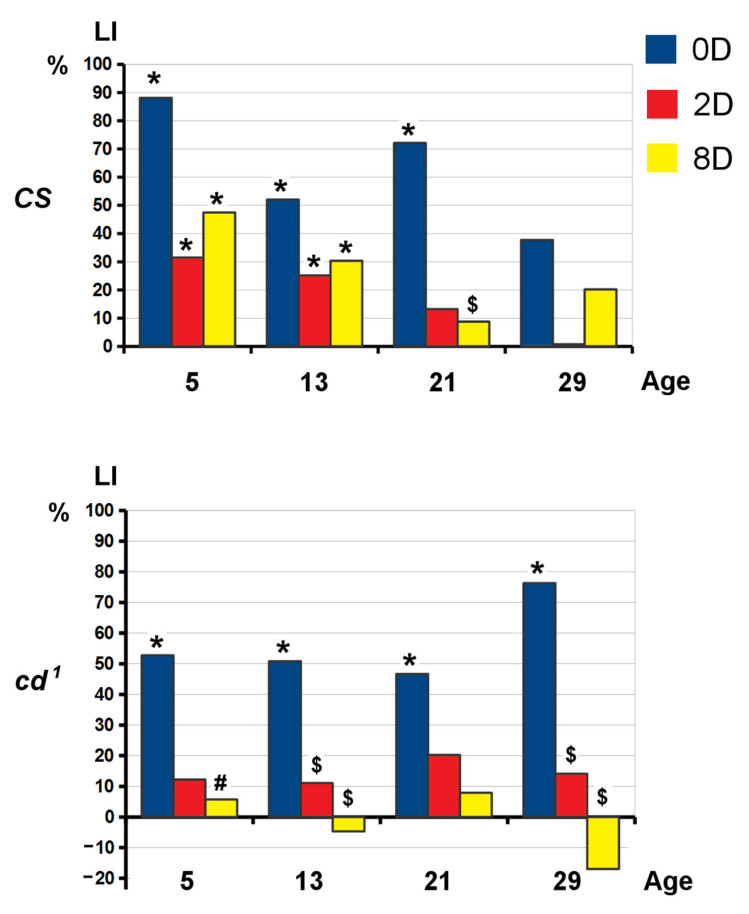
Learning and long-term memory retention in *CS* and *cd*^1^. *X* axis: the age where training was performed (days), *Y* axis: learning indices (LI), %. 0D—immediately after training (learning); 2D—2 days after training; 8D—8 days after training. Statistical differences: # from *CS*; $ from 0D, * from zero (two-sided randomization test; *p* < 0.05, n = 20).

**Figure 7 ijms-23-12356-f007:**
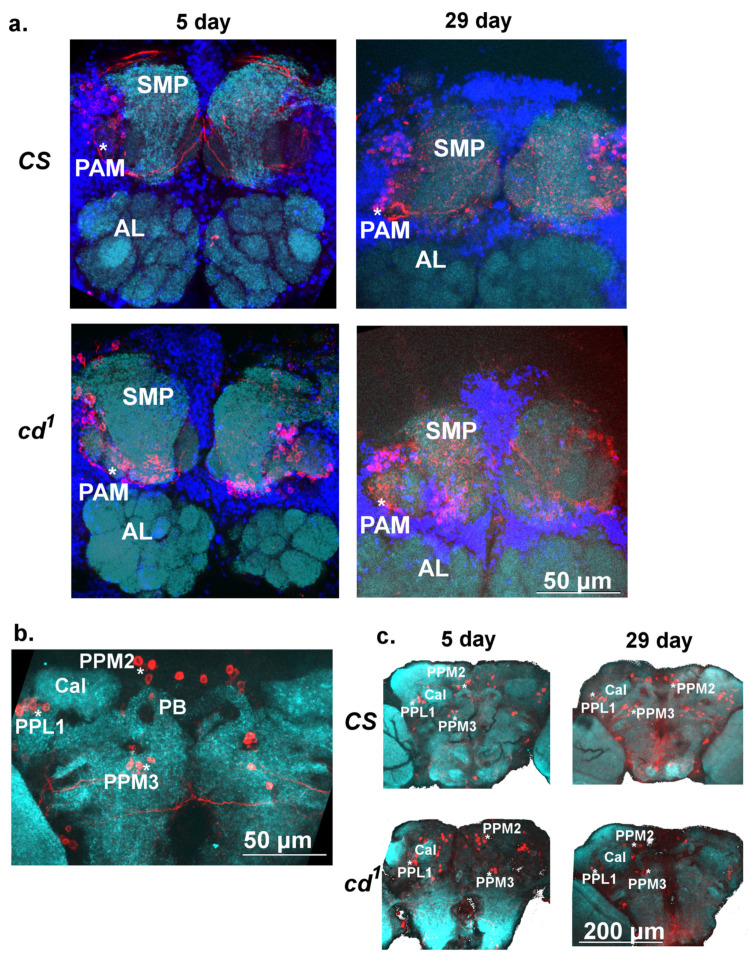
The tyrosine hydroxylase (TH)-positive clusters in the adult *CS* and *cd*^1^ brains. (**a**). PAM clusters of young (5 day-old) and aged (29 day-old) *CS* and *cd*^1^ (anterior side, X 63, scale bar 50 μm). (**b**). PPL1, PPM2 and PPM3 clusters of 5-day-old *CS* (posterior side, X 63, scale bar 50 μm). (**c**). PPL1, PPM2 and PPM3 clusters of young and aged *CS* and *cd*^1^ (X 20, scale bar 200 μm). Color scheme: cyan—neuropil (CSP); red—TH (DAN); blue—nuclei. AL—antennal lobe; Cal—calyx; PB—protocerebral bridge; SMP—superior medial protocerebrum. * TH-positive clusters. See abbreviations in the text.

**Figure 8 ijms-23-12356-f008:**
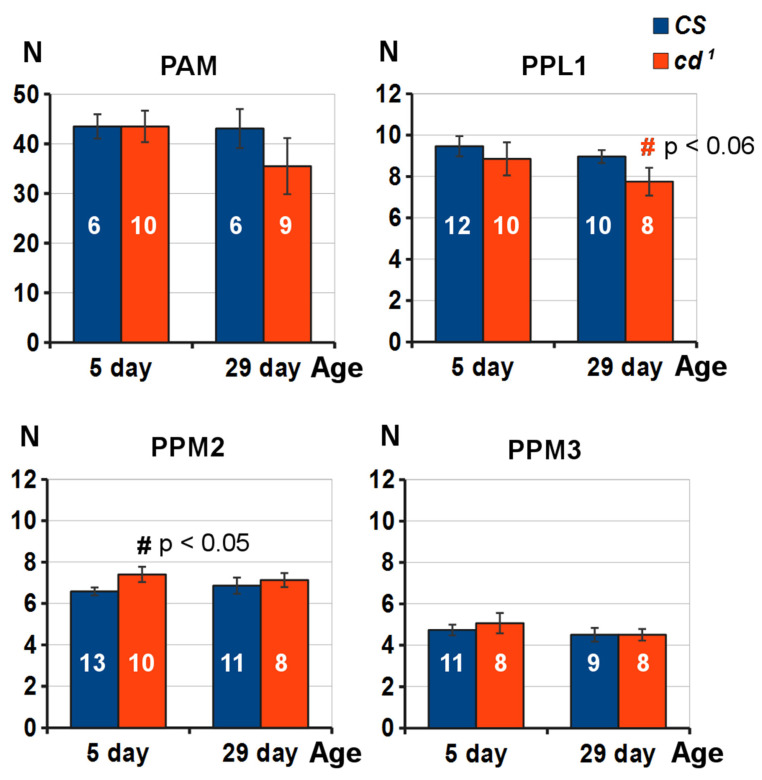
The average number of TH-positive neurons in the *Drosophila* DAN clusters. *X* axis: age of the fly (days); *Y* axis: cell number (N). Statistical difference: # from *CS*; two-sided randomization test, *p* is shown above the chart columns, n is marked with white numbers on the columns.

**Figure 9 ijms-23-12356-f009:**
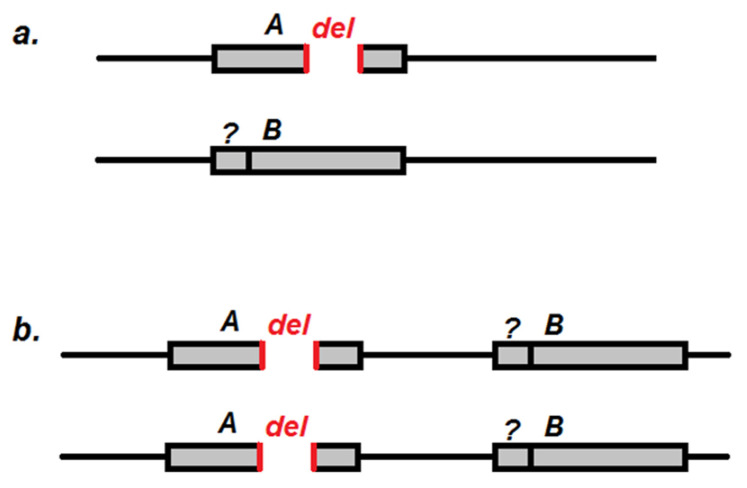
Possible arrangement of different *cd* copies in the genome of *cd*^1^ flies. Grey rectangle is either *cd* (case **a**) or *cd* followed by *Cyp6d4* (case **b**). *A* and *B*—two variants of *cd*; *del*—deletion; ?—the presumable alterations within *cd*(*B*) 5′ area.

**Figure 10 ijms-23-12356-f010:**
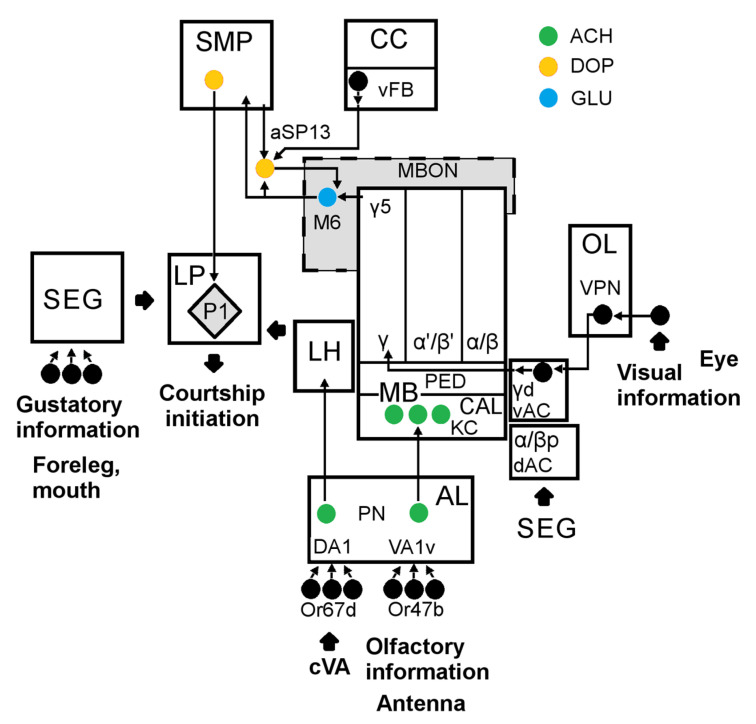
The simplified scheme of the *Drosophila* brain structures regulating courtship and memory formation in conditioned courtship suppression paradigm (CCSP). Abbreviations are given in text. Nerve cell bodies are shown as circles. Specific neurotransmitters are shown by colors.

## Data Availability

All data generated or analyzed during this study are included in this published article and its Appendix A.

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
