# Peer review of "cd1 Mutation in Drosophila Affects Phenoxazinone Synthase Catalytic Site and Impairs Long-Term Memory"

_ijms, 2022, doi:10.3390/ijms232012356_

Round 1

Reviewer 1 Report

The authors analyse a mutation on the cardinal (cd) that encodes a phenoxazinone synthetase involved in the kynurenine pathway, which leads to impaired long-term memory and dopaminergic neuron (DAN) alterations. Authors do the mapping and molecular characterization of the cd1 allele (following up on their previous study; Zhuravlev, 2018) and perform also the functional and phenotypic characterisation. The link of the kynurenine pathway to courtship behavior, nervous ystem and long-term memory is interesting also for a broader audience.

The study is interesting and well performed. However, the reviewer identifies a confusion in the naming of the gene. Authors name the gene phs and the protein PHS, while the mutant allele cd1. Few representative examples for the manuscript below:

"phs gene for phenoxazinone synthase (PHS) catalyzing 3-HOK dimerization has been presumed to harbor cd1 mutation"

"we have shown that cd1 phs is represented by at least two different copies"

"According to Flybase (www.flybase.org) phs location is 3R:22,694,979..22,697,990"

Based on Flybase the name of the gene is cardinal (cd): https://flybase.org/reports/FBgn0263986

Generally, it can happen that the gene name differs from that of the protein but it is quite unusual to use a different name for the gene vs. the alleles or mutations. Flybase mentions at the "symbol synonym" PHS as the name of the gene, based on the previous publication of the same group Zhuravlev et. al., 2018.

Moreover, in Flybase there is a gene called Phaser (Phs) with a different localisation (on 3L) with DNA-binding transcription factor activity: https://flybase.org/reports/FBgn0036522

Therefore, the suggestion here would be for the authors to call the gene cardinal (cd) and the mutant allele cd1 and thereby amend nomenclature throughout the manuscript. Thus, it would be different if the authors would wish to name the protein encoded by the cd gene PHS

Minor comments

1. authors should write correct micrograms as μg and not ug throughout the text (μ can be found in Symbols).

2. the phrase "Possible arrangement of different phs copies in cdgenome"  should be changed to "Possible arrangement of different cd allelic copies in the genome of cd1 flies". The term "cdgenome" is not really correct.

Author Response

Major comments:

Point 1: Therefore, the suggestion here would be for the authors to call the gene cardinal (cd) and the mutant allele cd1 and thereby amend nomenclature throughout the manuscript.

Response 1: The nomenclature was amended: “phs” was replaced by “cd”.

Minor comments:

Point 1: Authors should write correct micrograms as μg and not ug throughout the text (μ can be found in Symbols).

Response 1: Corrected.

Point 1: The phrase "Possible arrangement of different phs copies in cd1 genome" should be changed to "Possible arrangement of different cd allelic copies in the genome of cd1 flies". The term "cd1 genome" is not really correct.

Response 2: The phrase was corrected according to the reviewer's suggestion.

Reviewer 2 Report

Summary: This study describes experiments characterizing the nature of the cd1 Drosophila melanogaster strain, a mutant allele of the cardinal gene, the Drosophila homolog of phenoxazinone synthetase (PHS).  The authors show that cd1 mutants have two different molecular alterations in the cd gene – one containing a deletion and another with an unknown molecular impairment.  While cd1 mutants have altered enzyme activity, they have normal cd mRNA levels, but contain long-term memory formation defects.  The authors also show that cd1 mutants produce normal numbers of tyrosine hydroxylase (TH)-positive neurons in a relatively normal cellular distribution.  While the authors do go into a very in-depth description of their results and their Methods are very thorough, there are some concerns that I have for this study that are described below.

Major Comments

1)    One of my major concerns with this work is that the authors are basing their conclusions on the presence of two different molecular alterations of the cd gene in cd1 mutants on two flies.  How widespread is this phenomenon in this strain?  Can the deletion allele be homozygous or is it always heterozygous with the other molecular alteration?  Can you separate the two different alterations into unique stocks by crossing cd1 to a balancer chromosome?  I think that more evidence is necessary to support the conclusion that the cd1 strain contains two different molecular alterations (or alleles) of the cd gene.

2)    In lines 213-214, the authors describe how median courtship indices decrease with age and time after learning.  Were any statistics run on these data in Figure S3 to support this conclusion?  Statistical analysis needs to be included to support this conclusion.

3)    I find the data in Figure 6 (the long-term memory data) very hard to follow.  I think the color changes are very confusing; is there a better way to depict these data and any statistical differences?  I also think that the memory data from the cn1 mutants is extraneous and make the figure more complicated than it needs to be.  The rest of the manuscript focuses on the differences between CS control flies and cd1 mutant flies, so I think that this figure should also just include data from CS and cd1 mutant flies.  I think removing the cn1 data will help make the figure easier to follow.

4)    The Discussion spends a very long time describing the mechanisms of courtship and memory formation in CCSP; however, I think this is unnecessary for this manuscript.  This manuscript focuses on the molecular nature of the cd1 mutant strain and some characterization of long-term memory formation, so I think that the information in the Discussion from lines 331-400 is extraneous and should be removed.  That will allow the reader to focus on discussion of information more directly relevant to the data being described in this manuscript.

Minor Comments

1)    In the Introduction, the authors mention a number of eye color genes, but I think they are referring to mutants in these genes (i.e., lines 102, 104 and 114).  If the authors are referring to mutants in this section, they should include the word “mutants” after they list a specific gene.

2)    In line 191 the authors refer to Figure S3 and I think they mean to refer to Figure S2.

3)    For the graphs in Figures 5, 6 and 8, the authors should include the axis titles on the graphs themselves, not in the figure legends.  This will increase the clarity of the figures.

4)    In line 332 the authors refer to Figure 9 and I think they mean to refer to Figure 10.

5)    In general, I think the authors should proofread the entire manuscript for correct grammar.  There are several instances where articles or prepositions (i.e., the, an, etc.) are left out of the text.

Author Response

Major Comments

Point 1: One of my major concerns with this work is that the authors are basing their conclusions on the presence of two different molecular alterations of the cd gene in cd1 mutants on two flies. How widespread is this phenomenon in this strain? Can the deletion allele be homozygous or is it always heterozygous with the other molecular alteration? Can you separate the two different alterations into unique stocks by crossing cd1 to a balancer chromosome? I think that more evidence is necessary to support the conclusion that the cd1 strain contains two different molecular alterations (or alleles) of the cd gene.

Response 1: This is a really interesting question. At present, it is clear that cd1 carries the both variants of cd gene, with and without deletion, as sequencing shows A and B. None of them is fully active (as cd1 eye color differs from that of the wild type). We did not observe any phenotypic segregation and spontaneous recovery of the wild type eye color in cd1 population. As we mention in Discussion, two different cases are possible: 1. A and B are at the same locus; 2. they are at different loci in the genome. In our further research we plan to perform in situ hybridization to visualize the deleted fragment of cd gene on the cd1 polytene chromosomes to check whether it resides only in half of chromatids at cd locus (case 1), or in all chromatids but at some different loci (case 2). Isolation of different stocks and sequencing is also possible in future.

Point 2: In lines 213-214, the authors describe how median courtship indices decrease with age and time after learning. Were any statistics run on these data in Figure S3 to support this conclusion? Statistical analysis needs to be included to support this conclusion.

Response 2: We have placed the data on statistical analysis in Figure S3.

Point 3: I find the data in Figure 6 (the long-term memory data) very hard to follow. I think the color changes are very confusing; is there a better way to depict these data and any statistical differences? I also think that the memory data from the cn1 mutants is extraneous and make the figure more complicated than it needs to be. The rest of the manuscript focuses on the differences between CS control flies and cd1 mutant flies, so I think that this figure should also just include data from CS and cd1 mutant flies. I think removing the cn1 data will help make the figure easier to follow.

Response 3: We have omitted the color changes. All data regarding cn LTM were placed in Supplementary materials (Figure S4).

Point 4: The Discussion spends a very long time describing the mechanisms of courtship and memory formation in CCSP; however, I think this is unnecessary for this manuscript. This manuscript focuses on the molecular nature of the cd1 mutant strain and some characterization of long-term memory formation, so I think that the information in the Discussion from lines 331-400 is extraneous and should be removed. That will allow the reader to focus on discussion of information more directly relevant to the data being described in this manuscript.

Response 4: We have shortened and restructured this part of Discussion with the corresponding references. However, it cannot be totally removed, because it contains information crucial for understanding the possible mechanisms of affecting the courtship memory by KP mutations .

Minor Comments

Point 1: In the Introduction, the authors mention a number of eye color genes, but I think they are referring to mutants in these genes (i.e., lines 102, 104 and 114). If the authors are referring to mutants in this section, they should include the word “mutants” after they list a specific gene.

Response 1: Corrected.

Point 2: In line 191 the authors refer to Figure S3 and I think they mean to refer to Figure S2.

Response 2: Corrected.

Point 3: For the graphs in Figures 5, 6 and 8, the authors should include the axis titles on the graphs themselves, not in the figure legends. This will increase the clarity of the figures.

Response 3: The axis titles were included.

Point 4: In line 332 the authors refer to Figure 9 and I think they mean to refer to Figure 10.

Response 4: Corrected.

Point 5: In general, I think the authors should proofread the entire manuscript for correct grammar. There are several instances where articles or prepositions (i.e., the, an, etc.) are left out of the text.

Response 5: Corrected.

Reviewer 3 Report

The manuscript by Zhuravlev et al. concerned the role of cd1 mutation in long-term memory and molecular mechanisms of phs gene (damaged by cd1 mutation) action in Drosophila. The authors used wide range of up-to-date methods to study the structure and function of phs in a mutant cd1 strain in comparison with wild type flies and suggest a good scheme of the Drosophila brain structures regulating courtship and memory formation.

Overall, the manuscript by Zhuravlev et al. is a well written paper, the study described is well-designed, well-executed, and contains new data which are well discussed. The manuscript certainly is of interest to the readership of IJMS.

However, I have some questions and comments to the authors:

Lines 107-109. It seems that a reference on the data mentioned is absent. Please, provide it.

Lines 264-268. What age Drosophila were used in this experiment? The authors compared their data with the data by Mao and Davis and see the difference in the amount of dopaminergic neuronsns. But the dopamine level is known to change in Drosophila imago with age, and that is one possible explanation for the difference observed. Mao and Davis used 4-dyas-old flies, but I never found the information concerning the age of flies used in the MS by Zhuravlev et al.

Lines 486-487. Please, point out the wave length for DNA concentration measurements.

Lines 529-539. How many biological replicates were made? Please, provide the information.

Line 572. As I mentioned above, the age of flies used in this experiment need to be point out.

Lines 578-579. Why such a long incubation with first and, especially, second antibodies was used (3 and 2 days, correspondingly, instead of usual overnight)?

Minor comments:

Lines 479-480, 530. Please, change “ul” for “µl”.

Lines 504, 532, 580. Please, change “ug” for “µg”

Line 576. Please, add “(courtesy of Prof. Buchner, Germany)” after mention of corresponding antibodies.

Author Response

Major comments:

Point 1: Lines 107-109. It seems that a reference on the data mentioned is absent. Please, provide it.

Response 1: The reference was provided.

Point 2: Lines 264-268. What age Drosophila were used in this experiment? The authors compared their data with the data by Mao and Davis and see the difference in the amount of dopaminergic neuronsns. But the dopamine level is known to change in Drosophila imago with age, and that is one possible explanation for the difference observed. Mao and Davis used 4-dyas-old flies, but I never found the information concerning the age of flies used in the MS by Zhuravlev et al.

Response 2: We studied DAN distribution on days 5 and 29 (see line 245 in initial Manuscript). We have also specified the flies ages after lines 264-268 «on days 5 – 29 of Drosophila adult life” and specified their ages at the appropriate places of the manuscript, including Materials and Methods.

Point 3: Lines 486-487. Please, point out the wave length for DNA concentration measurements.

Response 3: Corrected: “at 260 nm”.

Point 4: Lines 529-539. How many biological replicates were made? Please, provide the information.

Response 4: Corrected: Three biological replicates were made in each case”.

Point 5: Line 572. As I mentioned above, the age of flies used in this experiment need to be point out.

Response 5: Corrected.

Point 6: Lines 578-579. Why such a long incubation with first and, especially, second antibodies was used (3 and 2 days, correspondingly, instead of usual overnight)?

Response 6: It was done to improve the brains staining. Similar long period of incubation was also used in our previous investigations (for example, see ref. 29, doi: 10.3389/fphys.2020.00971).

Minor comments:

Point 1: Lines 479-480, 530. Please, change “ul” for “µl”.

Point 2: Lines 504, 532, 580. Please, change “ug” for “µg”

Point 3: Line 576. Please, add “(courtesy of Prof. Buchner, Germany)” after mention of corresponding antibodies.

Response 1-3: All the suggested changes were made.

Round 2

Reviewer 2 Report

I thank the authors for addressing my comments.  I especially thank them for giving their thoughts on my first point about the nature of the cd1 mutation and the stock.  I think that the future experiments that they propose are good ones ("In our further research we plan to perform in situ hybridization to visualize the deleted fragment of cd gene on the cd1 polytene chromosomes to check whether it resides only in half of chromatids at cd locus (case 1), or in all chromatids but at some different loci (case 2). Isolation of different stocks and sequencing is also possible in future.") and it would add to the manuscript to include this text in the Discussion (perhaps in the paragraph located at lines 302-310).  Otherwise, I am satisfied with the manuscript.

Author Response

Point 1: I think that the future experiments that they propose are good ones ("In our further research we plan to perform in situ hybridization to visualize the deleted fragment of cd gene on the cd1 polytene chromosomes to check whether it resides only in half of chromatids at cd locus (case 1), or in all chromatids but at some different loci (case 2). Isolation of different stocks and sequencing is also possible in future.") and it would add to the manuscript to include this text in the Discussion (perhaps in the paragraph located at lines 302-310). Otherwise, I am satisfied with the manuscript.

Response 1: The text was included in the Discussion, according to the reviewer's suggestion. The possible source of the cd(B) gene copy is discussed as well.